# A Novel Ultrasound-Guided Cervical Plexus Block: A Cadaveric Canine Study

**DOI:** 10.3390/ani14213094

**Published:** 2024-10-26

**Authors:** Ariel Cañón Pérez, José I. Redondo García, Eva Z. Hernández Magaña, Agustín Martínez Albiñana, María de los Reyes Marti-Scharhausen Sánchez, Cristina Bonastre Ráfales, Pablo E. Otero, Ana García Fernández, Jaime Viscasillas

**Affiliations:** 1Department of Animal Pathology, Faculty of Veterinary Medicine, University of Zaragoza, C/Miguel Servet 177, 50013 Zaragoza, Spain; cbonastr@unizar.es; 2Experimental Surgery Unit (ESU), Vall d’Hebron Institut de Recerca (VHIR), Vall d’Hebron, Hospital Universitari, Vall d’Hebron Barcelona Hospital Campus, Passeig Vall d’Hebron 119-129, 08035 Barcelona, Spain; 3Departamento de Medicina y Cirugía Animal, Universidad CEU Cardenal Herrera, c/Tirant lo Blanc 7, Alfara del Patriarca, 46115 Valencia, Spain; nacho@uchceu.es (J.I.R.G.); eva.hernandezmagana@uchceu.es (E.Z.H.M.); 4Hospital Veterinario AniCura Aitana, c/Xirivella 16, Mislata, 46920 Valencia, Spain; agustin.martinez@anicura.es; 5Hospital AniCura Indautxu, San Mames Zumarkalea 36-38, 48010 Bilbo, Spain; reyesvet92@gmail.com; 6Instituto Universitario de Investigación Mixto Agroalimentario de Aragón (IA2), University of Zaragoza, C/Miguel Servet 177, 50013 Zaragoza, Spain; 7Department of Anesthesiology and Pain Management, Facultad de Ciencias Veterinarias, Universidad de Buenos Aires, Buenos Aires C1417DSE, Argentina; potero@fvet.uba.ar; 8London Vet Specialists, 56 Belsize Lane, Hampstead, London MW3 5AR, UK; ana.g.fdez@hotmail.com; 9Hospital Veterinario AniCura Valencia Sur, Avda, Picassent 28, 46460 Silla, Spain

**Keywords:** cervical nerves, cervical plexus block, dog, interfascial block, regional anesthesia

## Abstract

The cervical plexus block is commonly used in human medicine to provide analgesia to cervical structures, including dermatomes, lymph nodes, the larynx, the thyroid, parathyroid glands, and carotid vessels. This practice may also hold potential clinical value in veterinary medicine, particularly for dogs, where such a technique has not been previously documented. This study introduces a novel approach for performing a simple ultrasound-guided injection into the cervical plexus plane in dogs. The research involved exploring the neck anatomy of dogs, administering an injection of a contrast-dye solution, and assessing the distribution of the injected solution in six dog cadavers through both imaging scans and anatomical dissections. The injection was precisely delivered into the interfascial plane where the cervical plexus is located. The results demonstrated that the injections successfully targeted the C2 and C3 nerves in most cases; however, a spread to the C4 and C5 nerves was infrequent. There was no significant difference observed between the groups studied, and no contrast was detected in the epidural space. These findings suggest that while this technique is effective for targeting the C2 and C3 nerves, it is less reliable for reaching the C4 or C5 nerves. Future research involving live animals is warranted to further evaluate the effectiveness and safety of this technique in clinical settings.

## 1. Introduction

Interfascial plane blocks have proven effective as part of multimodal analgesia in various scenarios within veterinary medicine, encompassing soft tissue surgeries and orthopedic procedures [1,2,3,4]. One of these techniques, the ultrasound-guided cervical plexus plane (CPP) block, is suitable for preventing nociception and promoting post-surgical analgesia in humans [5]. The CPP block is recommended for providing analgesia to cervical structures, including cervical dermatomes, lymph nodes, the larynx, the thyroid, parathyroid glands, and carotid vessels [6,7].

In dogs, the ventral cervical plexus is composed of ventral branches originating from the first (C1) to the fourth (C4) and sometimes the fifth (C5) cervical spinal nerves [8]. These branches traverse the interfascial plane between the omotransversarius muscle and the deep muscles of the neck, surrounded by the deep cervical fascia [9]. The omotransversarius muscle is a longitudinal and substantial muscular band that is flattened on the lateral side of the neck and extends from the shoulder surface to the atlantoidea region. The ventral branch of C1 is connected to the vagosympathetic trunk and the accessory nerve, and it does not provide cutaneous innervation. The ventral branch of C2 divides into the great auricular and transverse cervical nerves, which are responsible for innervating the parotid region, the base of the external ear, the skin of the convex face of the auricle, as well as the laryngeal region and intermandibular space, respectively. The ventral branches of C3–C5 innervate the hypaxial cervical muscles and provide sensory innervation to the soft tissues in the region. The dermatomes in the proximal half of the neck’s lateroventral region are innervated by the cutaneous branches of the ventral branches of the C2 and C3 cervical nerves (Figure 1).

The cervical fascia is an extensive fibrous system divided into several layers that form natural sheaths around cervical structures. These layers are especially well differentiated in the ventral region of the neck, allowing for the identification of three distinct components: the superficial lamina, the pretracheal lamina, and the prevertebral lamina, each contributing uniquely to the anatomical structure [9]. Within this complex fascial arrangement, the ventral branches of the cervical nerves are enclosed, highlighting their anatomical integration within fascia. Therefore, this interfascial plane could be a suitable location for administering local anesthetics to block these nerves. Although there have been reports of blocking the ventral branches of cervical spinal nerves in veterinary medicine, such as in horses where a technique was described and its efficacy evaluated [10,11], a case of the use of a cervical block in a dog [12], a technique for an ultrasound-guided interfascial plane block to selectively desensitize the cervical plexus in dogs, has not been described yet.

Therefore, the purpose of this study was twofold: (1) to describe the ultrasound anatomy of the lateral aspect of the cervical cranial region and evaluate the feasibility of performing an ultrasound-guided cervical plexus plane (US-CPP) injection in canine cadavers, and (2) to examine the distribution pattern and nerve staining of a dye-contrast solution injected into the interfascial plane between the omotransversarius muscle and the deep muscles of the neck (i.e., the longus capitis and longus colli) using a gross anatomical dissection and CT-scan.

This study hypothesized that an ultrasound-guided craniocaudal approach could successfully perform a US-CPP by injecting a dye-contrast solution into the interfacial plane beneath the omotransversarius muscle. This method would selectively stain the ventral branches of the cervical spinal nerves from C2 to C4 in canine cadavers.

## 2. Materials and Methods

### 2.1. Animal Use and Euthanasia

A total of seven fresh canine cadavers, euthanized for reasons unrelated to the present study, were used, with all injections performed within 6 h post-euthanasia. At the time of the injections, signs of rigor mortis were absent. The cadavers included five mixed-breed dogs, one German Shepherd, and one Mastiff cross, which was initially used for the ultrasonographic recognition and anatomic study. The specimens with noticeable cervical abnormalities were excluded. The remaining six cadavers were used to evaluate the spread of the injectate and nerve staining following a US-CPP injection with a dye-iodinated contrast. The selection of cadavers was based on specimen availability, considering the descriptive design of this phase.

### 2.2. Study Phases

This study was divided into two phases. In Phase I, the sonoanatomy of the lateral aspect of the cervical region was described, and the US-CPP injection was tested. Additionally, the gross anatomy of the cervical region, including the cervical plexus, was assessed in one canine cadaver. Phase II evaluated the spread of the injectate and nerve staining following the US-CPP injection with a dye-iodinated contrast, using CT scanning and a gross anatomical dissection in the six cadavers.

#### 2.2.1. Phase I: Ultrasonographic Examination and Anatomic Study

In this phase, a female canine cadaver weighing 37 kg, with body condition scores of 3/5 was included.

Following the clipping of hair from the cervical regions, an ultrasound scan of the latero-ventral surface of the neck was performed to establish a relationship between the ultrasound and gross anatomic findings and to identify possible landmarks to guide the US-CPP injections. The cadavers were placed in lateral recumbency, with the side to be scanned placed uppermost, and ultrasound gel (Transonic Gel Clear, TELIC, Barcelona, Spain) was applied to enhance the acoustic coupling. An experienced anesthesiologist conducted the ultrasound study with a linear ultrasound probe (HFL38x/13-6, Sonosite, Bothell, WA, USA) attached to an ultrasound machine (either a Sonosite M-turbo or Sonosite X-Porte, WA, USA). The ultrasound probe was positioned under the transverse process of the axis, parallel to the longitudinal axis of the spine, with its mark facing cranially. Afterward, the probe was moved and positioned at various angles to locate the specific area of interest. In this phase, we simulated and determined the best needle access route for an in-plane approach. After completing the scanner, the procedure was repeated on the contralateral side.

Immediately after carrying out the ultrasound evaluation, an anatomical dissection was initiated. First, the cervical region’s skin was removed to expose the cervical fascia and muscles. Then, the platysma muscle was removed, and the cleidocervical and omotransversarius muscles were carefully detached from their insertion in the wing of the atlas and lifted to expose the ventral branches of the cervical spinal nerves from C1 to C6. Additionally, the accessory nerve and the phrenic nerve were localized and inspected.

#### 2.2.2. Phase II: Ultrasound-Guided CPP Injections, CT-Scans, and Gross Anatomical Dissections

In this phase, 12 cervical plexuses from the six canine cadavers, three males and three females, weighing 27 (7.1–55) kg [median (range)], with a body condition score of 3/6 (range 2–6), were included. A total volume of 0.15 mL kg–1 of a mixed solution of 1:5 methylene blue (PB.5297, Euromex, Genk, The Netherlands) and iopromide (Ultavist-300, Bayer, Barcelona, Spain) was injected bilaterally.

##### Ultrasound-Guided CPP Injection

After the hair removal from and cleansing of the cervical region, the cadavers were positioned in lateral recumbency with the region to be scanned uppermost. To enhance the visualization of the interfacial plane and provide conditions to guide the needle, a small cushion was placed under the neck.

Based on Phase I, the transducer was positioned parallel to the cervical spine, immediately caudal to the wing of the atlas and ventral to the axis (Figure 2). When the cleidocervical and omotransversarius muscles and the underlying interfascial plane containing the ventral branches of C2 and C3 were recognized, a prefilled 20-gauge Quincke spinal needle (Spinocan, BBraun, Barcelona, Spain), attached to a syringe containing the dye solution, was introduced in a craniocaudal direction using an in-plane technique under direct ultrasound visualization until the needle tip reached the interfacial plane caudal to the ventral branch of C2. When satisfactory needle positioning was achieved, 0.15 mL kg^−1^ of the dye solution was injected. After completing the injections at the target site, the cadaver was repositioned to inject on the contralateral side. The same person carried out all the techniques.

##### CT-Scan

Immediately after completing the US-CPP injections, a CT-scan was performed with the cadavers positioned in lateral recumbency. All studies were performed using a 16-slice scanner (Brivo, General Electric, Milwaukee, WI, USA), and the images obtained had a slice thickness of 0.625 mm. All images were visualized in the bone algorithm, where the transverse and sagittal slices were reviewed. Two persons (a board-certified radiologist and an anesthesiologist) evaluated the images independently to determine the iodinated craniocaudal longitudinal spread, proximity to the esophagus and trachea, and spread to the epidural space.

##### Gross Anatomical Dissection

Immediately after finishing the CT-scans, a bilateral dissection of the cervical region was performed to evaluate the spread of dye and nerve staining, following the technique described in Phase I, by the same researcher assigned to perform the injections. The nerves were considered successfully stained if their entire circumference was stained over a distance of at least 0.6 cm [13]. Any complications associated with the technique, such as evidence of needle advancement within the carotidal or tracheal interfascial spaces or spread of dye to non-target planes or nerve branches, were also assessed and recorded.

### 2.3. Statistics

Descriptive statistics were computed for both the dye (“INK”)- and iodinated-contrast (“CONTR”) techniques, including frequency and spatial distribution. “INK” refers to the staining of the ventral cervical nerve roots, while “CONTR” represents the distribution of the iodinated contrast relative to the vertebral bodies. A Chi-Square test was employed to evaluate the differences in contrast distribution between the two techniques. Differences were considered significant if *p* < 0.05.

## 3. Results

### 3.1. Phase I. Ultrasonographic Examination and Anatomic Study

The ultrasonography of the lateral region of the neck at the level of the ventral aspect of the axis allowed for the differentiation of the bellies of the muscles, the interfascial planes between them, and the bones structures. The wing of the atlas was recognized as a rounded hyperechoic line that produced an evident acoustic shadow, which was also generated by the transverse processes and vertebral bodies of the caudal cervical vertebrae when exploring the region. From lateral to medial, two hypoechoic structures were visualized, corresponding to the cleidocervical and the omotransversarius muscles. Below the omotransversarius muscle, the interfascial plane containing the ventral branches of the cervical spinal nerves was visualized as hyperechoic. When the transducer was located caudal to the wing of the atlas, the ventral branches of C2 and C3 were observed as two hyperechoic ellipsoidal structures with an anechoic center (Figure 2).

The gross anatomical dissections correlated with the ultrasound findings (Figure 3). Both methods allowed for the identification of the interfascial plane and the ventral branches of C2 and C3 (Appendix A).

### 3.2. Phase II Ultrasound-Guided CPP Injections, CT-Scans, and Gross Anatomical Dissections

#### 3.2.1. Ultrasound-Guided CPP Injection

Ultrasound scanning allowed for the identification of the landmarks and injection sites in all the scanned regions. The consistent location of the target interfascial plane was observed in all the necks studied. Injectate distribution in the target fascial plane was visualized for all the injections.

#### 3.2.2. CT-Scan

The cranial margin of the iodinated contrast solution reached the C1, C2, and C3 vertebral bodies in 3/12, 11/12, and 12/12 injections, respectively. The caudal margin of the contrast reached the C4, C5, and C6 vertebral bodies in 8/12, 5/12, and 1/12 injections, respectively (Figure 4).

No contrast was observed in the epidural space. The iodinated contrast was found at <5 mm from the esophagus and the trachea in 2 out of the 12 cases (Figure 5).

#### 3.2.3. Gross Anatomical Dissection

All injections were found in the interfascial plane connected to the cervical plexus. The anatomical dissections revealed that the US-CPP injection resulted in a craniocaudal dye dispersion from the injection site, which predominantly stained the ventral branches of the cervical spinal nerves C2 and C3. The ventral branches of the C1, C2, C3, C4, and C5 nerves were stained in 3/12, 10/12, 8/12, 2/12, and 0/12 injections (Figure 6), respectively.

The results indicate that the accessory nerve, located between the cleidocervical and omotransversarius muscles, and the phrenic nerve remained unstained in all the specimens.

The Chi-Square test revealed no significant difference in the distribution of the injected mixture between the ‘INK’ and ‘CONTR’ techniques (*p* > 0.05). This result indicates that both the staining and iodinated contrast methods demonstrated comparable distribution patterns in our cadaveric model. Therefore, the selection of either technique may not substantially influence the observed distribution outcomes.

The distribution of the injected contrast and nerve staining is graphically summarized in Figure 7. 

## 4. Discussion

This study findings demonstrate the feasibility of ultrasound-guided identification of the interfascial plane housing the ventral branches of the cervical plexus nerves in dogs. Additionally, the injection technique employed proved highly effective, achieving interfacial dye distribution in six out of six of the cases tested.

The CT images showed that the distribution of the contrast reached the C1 to C2 vertebrae cranially and extended caudally from the C3 to C5 vertebrae, with only one case reaching the C6 vertebra. The margins of the contrast distribution were irregular, consistent with findings from other cadaveric studies of interfascial injections in dogs [14,15] and in humans [16]. This irregularity may be attributed to several factors. Firstly, it could be associated with the complexity of the cervical fascia anatomy, which extends from the subcutaneous tissue to the insertions on the hyoid bones, atlas, ribs, and sternum, and encompasses muscles as well as other structures such as the trachea, esophagus, thyroid glands, and major vessels [9]. Secondly, the injection site, located at the intersection between the superficial and deep cervical fascia, may allow slight variations in injection depth, resulting in a distribution of greater or lesser depth and potentially reaching structures such as the esophagus or trachea, as observed in some cases. Thirdly, the considerable racial pleomorphism could be another cause of the distribution differences found; however, in our study, the predominance of mixed breeds limited the assessment of the specific variations between breeds. Further studies are needed to assess whether the point of injection or the animal’s anatomy influences contrast distribution.

Interfascial dye distribution resulted in the staining of the ventral roots according to the results obtained by the authors. Thus, while the C2 and C3 nerves were stained in most injections, the C1 and C4 nerves were stained in only a few cases. However, the selective absence of C1 nerve staining is desirable, as it preserves the nerve’s exclusive motor function. The low staining rates for the C4 and C5 nerves may be due to the low volume injected. The initial injection volume (0.15 mL/kg) was chosen based on the volumes previously reported for interfascial blocks in dogs [17,18,19]. While this volume was adequate for staining the C2 and C3 nerves, it was insufficient for staining the C4 nerve. An increase in the administered volume has been a factor influencing the extent of contrast spread in fascial blocks [20,21]. However, other studies have not shown the same results, as increasing the volume did not extend the spread [22,23,24]. Further studies are needed to determine if adjusting the injection volume can impact the distribution pattern in this technique.

In human medicine, a cervical plexus block has been used for approximately a century, describing different approaches according to the depth at which the injections are made. There are three blocking locations: the deep or paravertebral block, the intermediate block (performed by injecting the local anesthetic agent into the cervical interfascial space, similar to the present study’s approach), and the superficial block (performed through a subcutaneous injection around the superficial branches) [5]. The intermediate cervical block approach could be a tool to reduce or avoid the side effects of performing a deep cervical plexus block (including impaired diaphragmatic movement, glossopharyngeal and/or hypoglossal nerve palsy, and an intrathecal injection, among others) [25]. Indications for this block include analgesia for a thyroidectomy, lymph node removal, a cervical node biopsy, and cervical and neck neuralgia, among others, with these being the most common indications of a carotid endarterectomy [6,7]. In dogs, based on their anatomy and our findings, the possible indications for a cervical blockage could be some of those listed above for humans, such as surgery of the thyroid or parathyroid gland, and a lymph node biopsy or masses in the cervical region of reference, among other possible indications such as surgical procedures for laryngeal hemiplegia.

The nociceptive innervation of the neck is carried out by the dorsal and ventral branches of the C2, C3, and C4 nerves; the ventral branches of C2 and C3 are responsible for the sensory innervation of the lateroventral surfaces of the cranial half of the neck. The description of the innervation of the dermatomes is well defined in the literature; there is an extensive overlap of the adjacent areas of innervation with the different cutaneous branches [26], which should be considered when planning the area that will be involved in the surgery.

Reports of regional anesthesia techniques with a similar approach in veterinary medicine have been published on horses undergoing surgery for laryngeal paralysis. They have proven effective at decreasing anesthetic requirements without significantly increasing pre-surgical times and without the reported complications derived from a blockage [11]. In another study, better surgical conditions were reported compared to a local anesthetic’s infiltration in the surgical area [10]. Finally, using this ultrasound-guided technique in addition to an incision line block provided adequate perioperative analgesia for left arytenoid lateralization in a dog without reporting complications [12].

The potential complications of regional anesthesia to consider are vascular punctures or an intravascular injection of an anesthetic agent; those risks could be reduced by performing ultrasound-guided techniques [27]. Ultrasonography can also reduce the risk of other complications, such as an inadvertent subarachnoid or epidural injection. Due to its anatomical complexity, the distribution of the injected volume in the cervical area could affect structures such as the vertebral canal, phrenic nerve, vagosympathetic trunk, and recurrent laryngeal nerve, among other structures. Neuraxial spread can be potentially avoided by administering a low volume of local anesthetics as suggested by the results of Stundner [28]. In human medicine, a potential phrenic nerve block has been described [29]. This nerve originates in humans from the ventral roots of the C3, C4, and C5 nerves; the block will affect some of these roots.

However, in dogs, the phrenic nerve originates from the ventral roots of the C5, C6, and C7 nerves, with an inconstant contribution from the C4 nerve [30]. Therefore, the risk of affecting this nerve would be considerably lower in dogs compared to humans. No phrenic nerve involvement was observed during the dissections in this study. Except for two cases where the C4 ventral branch was stained, the phrenic nerve’s origin remained unaffected, which may support this assumption. Although this complication has been described for other locoregional anesthesia techniques, such as paravertebral brachial plexus blocks in dogs [31], it is important to note that we cannot entirely rule out the possibility of bilateral phrenic nerve involvement with unilateral inoculation; however, based on our findings, this is considered unlikely. Additionally, a unilateral phrenic nerve block has been reported to have a minimal impact on ventilation in healthy, awake dogs [32].

This study has some limitations. First, when using this contrast mixture (iopromide and methylene blue) to assess its distribution, the viscosity and density differ from the local anesthetics. Therefore, there might be differences in the distribution of our mixture compared to local anesthetics. Furthermore, the distribution of the mixture in cadavers can also be slightly different to living animals. Finally, the anesthetist who performed all the injections was the same one who performed the dissections, so the procedures were not conducted in a blinded manner. Although this could be considered a bias, the assessment of the root staining was performed by two of the authors, and also the contrast distribution of the CT images correlated with the results found during the dissections.

## 5. Conclusions

The ventral branches of the C2 and C3 nerves can be stained in the interfascial plane by a single ultrasound-guided CPP injection technique, which could have potential clinical applications. The injected volume of 0.15 mL kg^−1^ rarely reaches C4 and C5. Clinical studies are needed to assess this technique’s analgesic efficacy and safety in vivo and determine its use.

## Figures and Tables

**Figure 1 animals-14-03094-f001:**
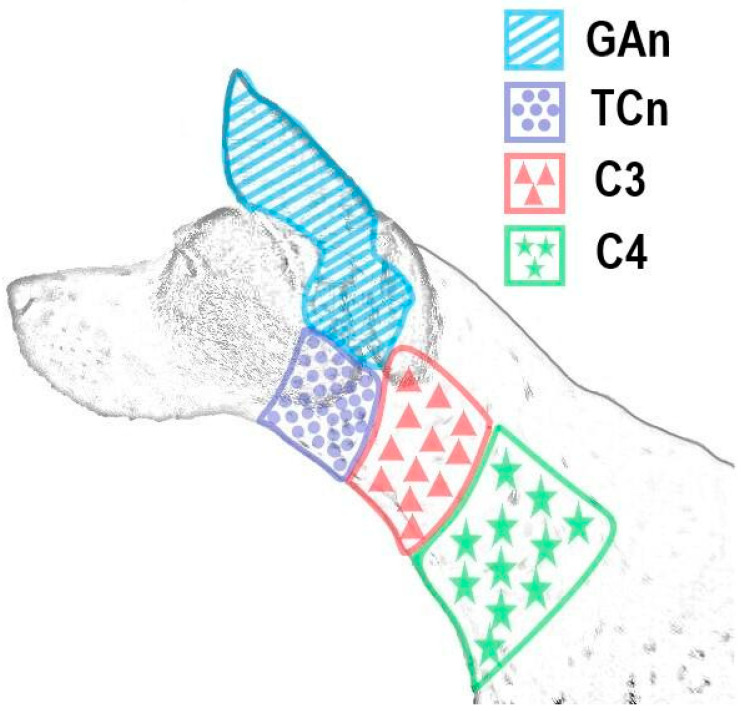
Dermatomal innervation of C2 to C4 ventral branches of cervical nerves. C3; C3 ventral cutaneous branch, C4; C4 ventral cutaneous branch, GAn; greater auricular nerve (C2 ventral branch), TCn; transverse cervical nerve (C2 ventral branch).

**Figure 2 animals-14-03094-f002:**
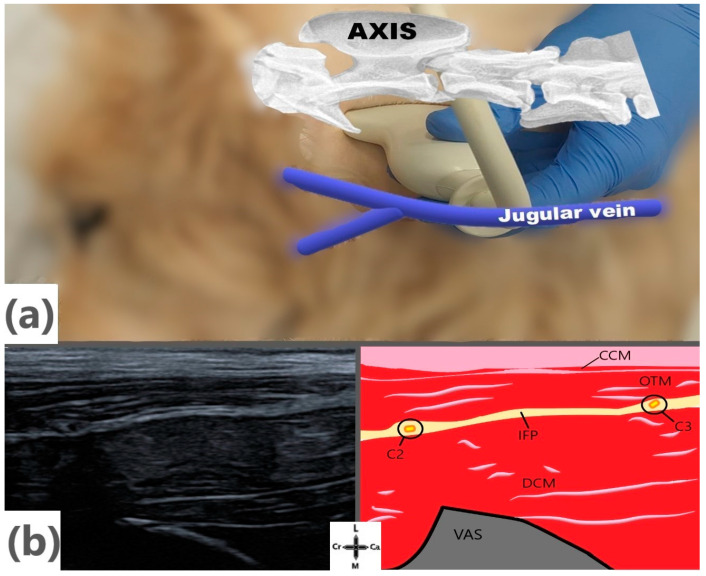
(**a**) Probe positioning perpendicular to the skin with exposed anatomical landmarks. (**b**) Acoustic window of desired injection site. C2; 2nd cervical nerve ventral branch, C3; 3rd cervical nerve ventral branch, Ca; caudal, CCM; cleidocervical muscle, Cr; cranial; DCM; deep cervical muscles, IFP; interfascial plane, L; lateral, M; medial, OTM; omotransversarius muscle, VAS; vertebral acoustic shadow.

**Figure 3 animals-14-03094-f003:**
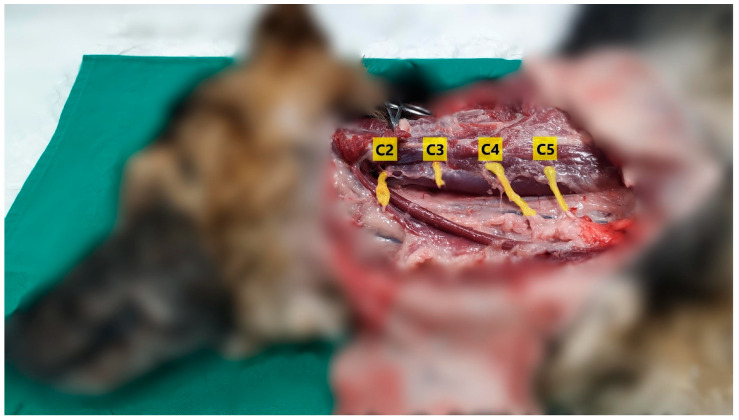
Anatomical dissection where the C2 to C5 ventral branches are yellow-highlighted.

**Figure 4 animals-14-03094-f004:**
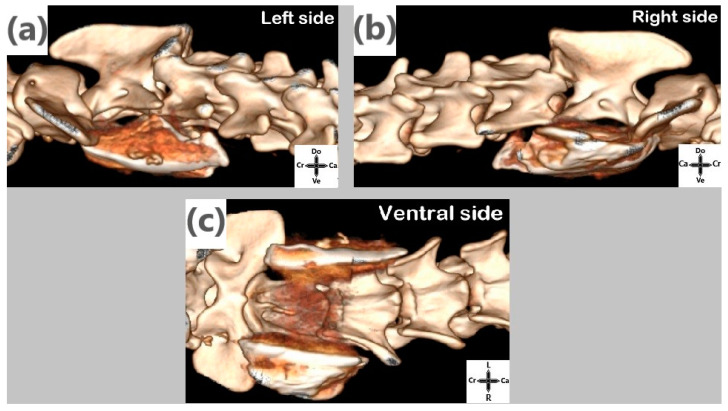
Computed Tomography 3D-reconstruction showing iodinated contrast spread. (**a**) Left-side spread reaching from C2 to C3–C4. (**b**) Right-side spread reaching from C2 to C3. (**c**) Ventral side showing both right and left spread. Ca; caudal, Cr; cranial, L; left, R; right.

**Figure 5 animals-14-03094-f005:**
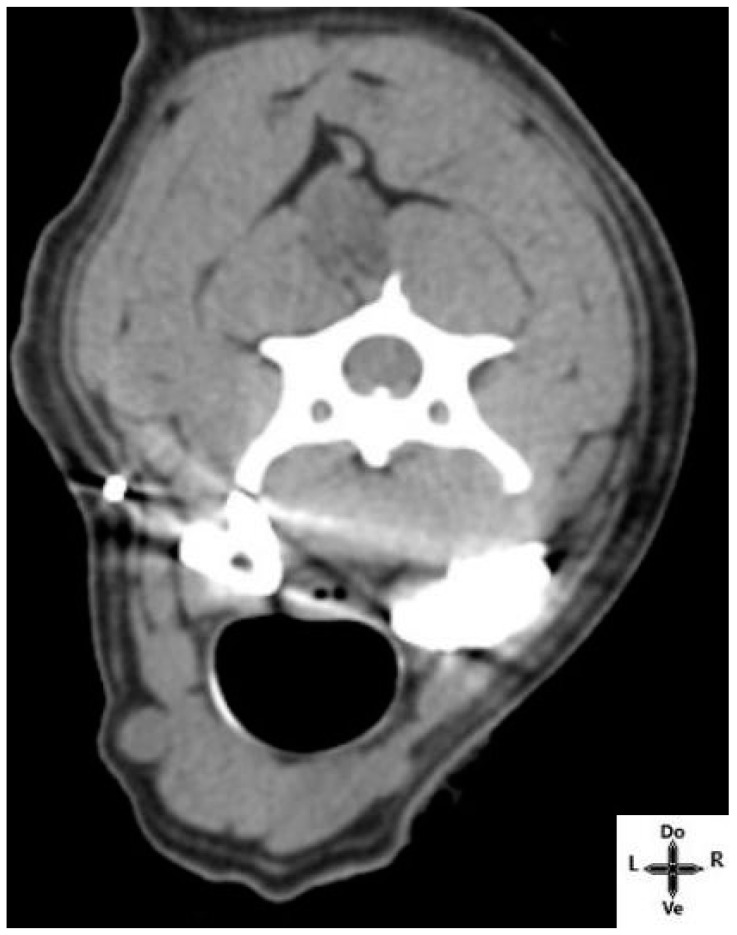
Axial Computed Tomography images after dye-contrast injections. Contrast spread to deeper structures (trachea and esophagus). Do; dorsal, L; left, R; right, Ve; ventral.

**Figure 6 animals-14-03094-f006:**
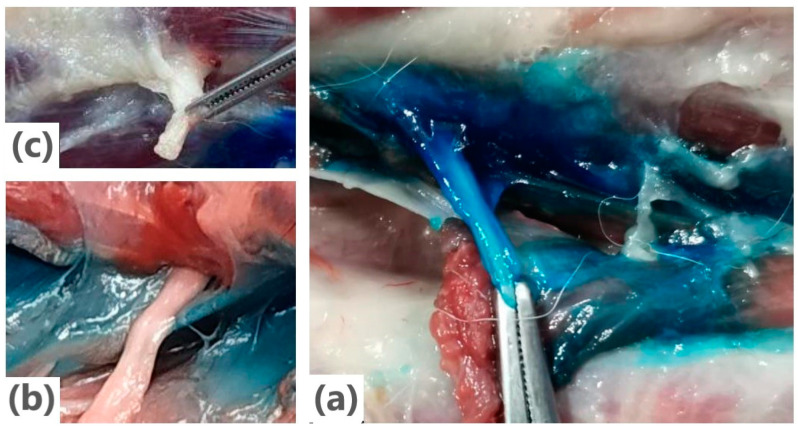
Criteria applied to assess staining. (**a**) Nerve stained. (**b**) Fascia stained; nerve not stained. (**c**) Nothing stained.

**Figure 7 animals-14-03094-f007:**
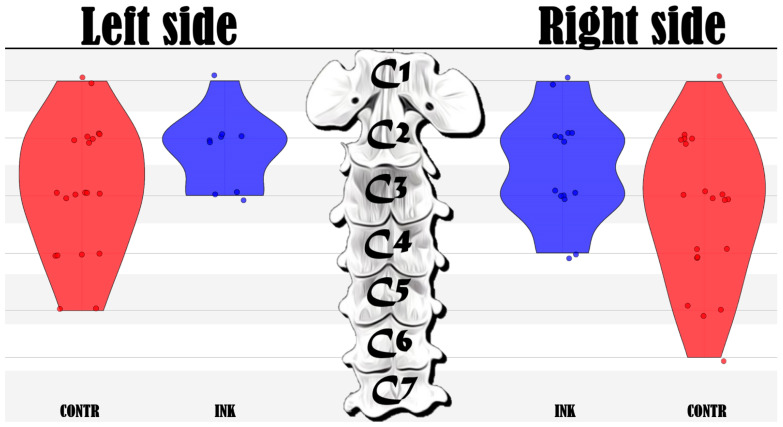
Violin plot and scatterplot of ink distribution evaluated through dissection (INK-Blue color) and contrast evaluated with a CT (CONTR-Red color) on the right and left sides of the six canine cadavers (12 injections).

## Data Availability

The complete data pertaining to this study are contained within the article.

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
