# Peer review of "A Novel Ultrasound-Guided Cervical Plexus Block: A Cadaveric Canine Study"

_animals, 2024, doi:10.3390/ani14213094_

Round 1
Reviewer 1 Report
Comments and Suggestions for Authors
Dear Authors, I have carefully reviewed the current manuscript.
The aim of this study was to describe a novel ultrasound-guided cervical plexus plane block in canine cadavers.
It seems an interesting idea on a clinically useful topic such as pain management, where regional anaesthesia is an integral part of multimodal opioid-free or opioid-sparing approaches, especially nowadays when the opioid crisis in humans may limit access of the veterinary community to opioids.
This block has been well described in humans; however, this approach has not been implemented in dogs under clinical conditions. It has been contributed to enhanced patient care by reducing systemic analgesics and providing effective pain management, and the current block seems to provide dense anaesthesia and analgesia in the distribution of C2-C4 nerve roots.
Your cadaveric study provided initial evidence that the distribution of contrast reached the C1 to C2 vertebrae cranially and extended caudally to the C3 to C5 vertebrae. In vivo studies should be designed to investigate the potential clinical application of that technique.
In my opinion this is a very well structured and designed work. I have no specific comments to make on the design, as it is a typical cadaveric study evaluating the regional block.
The introduction section has relevant references and the conclusions of the study address the main research questions.
Author Response
Comment 1: It seems an interesting idea on a clinically useful topic such as pain management, where regional anaesthesia is an integral part of multimodal opioid-free or opioid-sparing approaches, especially nowadays when the opioid crisis in humans may limit access of the veterinary community to opioids.
Response 1: We sincerely appreciate your valuable feedback and are pleased that you found our work interesting, particularly regarding the significance of regional anesthesia in multimodal protocols. Your comments reinforce the importance of continuing research in this area and enhancing our clinical practices.
Final comments: Your cadaveric study provided initial evidence that the distribution of contrast reached the C1 to C2 vertebrae cranially and extended caudally to the C3 to C5 vertebrae. In vivo studies should be designed to investigate the potential clinical application of that technique. In my opinion this is a very well structured and designed work. I have no specific comments to make on the design, as it is a typical cadaveric study evaluating the regional block. The introduction section has relevant references and the conclusions of the study address the main research questions.
Final comments response: We would like to sincerely thank you for your positive review and thoughtful evaluation of our work. Your encouraging feedback and the time you dedicated to assessing our research are greatly appreciated and motivate us in our ongoing efforts.
Reviewer 2 Report
Comments and Suggestions for Authors
Reviewer comments
1. It would be beneficial for the authors to clarify the meaning of "3/12," "11/12," and "12/12" in the abstract
2. It may be great for the authors to provide more explanation of the experimental groups mentioned in the abstract.
3. There appears to be a difference in font size (lines 61 and 62).
4. The authors should indicate the time of dye or contrast media injection post-euthanasia. This information would be useful in understanding any potential impact of rigor mortis.
5. It would be great for the author to include a reference for the dose of 0.15 ml kg-1 injection.
6. Would it be possible to divide the 'Materials and Methods' section into distinct subsections for clarity and ease of understanding? For example, Animal use, Euthanasia protocol, Injection protocol, Dissection protocol, CT Scan, and Approval of Animal Care Committees, etc. This would help provide a clear and organized presentation of the methodology.
Author Response
Comment 1: It would be beneficial for the authors to clarify the meaning of "3/12," "11/12," and "12/12" in the abstract
Response 1: Thank you for your suggestion. We have clarified the meaning of '3/12,' '11/12,' and '12/12' in the abstract as '3 out of 12 (3/12),' '11/12,' and '12/12' to enhance readability and ensure clarity for the readers.
Comment 2: It may be great for the authors to provide more explanation of the experimental groups mentioned in the abstract.
Response 2: Thank you for your valuable suggestion regarding the clarification of the experimental groups. We have revised the manuscript to include additional details about the canine cadavers used in the study. We hope that these changes meet your expectations and enhance the clarity of our experimental design.
Comment 3: There appears to be a difference in font size (lines 61 and 62).
Response 3: Thank you for bringing to our attention the mistake regarding the font size. We appreciate your careful review and have corrected this issue in the manuscript.
Comment 4: The authors should indicate the time of dye or contrast media injection post-euthanasia. This information would be useful in understanding any potential impact of rigor mortis.
Response 4: Thank you for your careful review and for highlighting the importance of including additional information in the article. We hope that the modification has correctly clarified this point.
Comment 5: It would be great for the author to include a reference for the dose of 0.15 ml kg-1 injection.
Response 5: Thank you very much for your contribution. We would like to clarify that in the following article cited in the bibliography, a volume of 0.15 ml kg-1 was utilized in one of the experimental groups:
- Garbin M, Portela DA, Bertolizio G, Garcia-Pereira F, Gallastegui A, Otero PE. Description of ultrasound-guided quadratus lumborum block technique and evaluation of injectate spread in canine cadavers. Vet Anaesth Analg. 2020 47(2), pp. 249–258.
Comment 6: Would it be possible to divide the 'Materials and Methods' section into distinct subsections for clarity and ease of understanding? For example, Animal use, Euthanasia protocol, Injection protocol, Dissection protocol, CT Scan, and Approval of Animal Care Committees, etc. This would help provide a clear and organized presentation of the methodology
Response 6: We appreciate the reviewer’s suggestion to divide the "Materials and Methods" section into subsections. This has been implemented for improved clarity.
Reviewer 3 Report
Comments and Suggestions for Authors
Congratulations.
The paper is well written in my opinion.
The introduction, apart from a few long sentences, is well structured and comprehensive.
Materials, methods and results are fairly well described. A few more efforts of completeness on the CT part (both for materials and methods as well as for results) and on the presentation of statistics would be appreciated (it does not compromise the quality of the paper).
The discussions are very interesting, comprehensive but not so long that the reader loses attention. Below you will find a word file with my brief remarks. They are more of my personal curiosities/thoughts to share with you and which I hope will give you new insights that you can also include in the discussions if you think it makes sense.
I am not qualified to correct English.
I thank you
Review: “A novel ultrasound-guided cervical plexus block. A cadaveric canine study”
Simple summary
Line 25-27: I suggest dividing the period into two sentences. You speak about human medicine before and then veterinary medicine in the same sentence. In this way I think it is clearer.
Abstract
Well written
Introduction
Line 67-71: I suggest dividing the sentence into two period. Like this the sentence is long
Material and methods
Line 143: “the platysma was removed”… I suggest to insert “platysma muscle…
Line 149-150: Do you find certain anatomical differences between cadavers? For example, different breeds or dolichomorph vs mesomorph vs brachimorph dogs?
Results
Well presented
Discussion
Line 275-276: In addition to your remarks on dissection (which I believe to be correct) I have another observation. It would be interesting to know if the solution arrived at C6 what type of dog breed it was. And it would also be interesting to know if the orientation of the spinal needle. Actually this is my personal impression, a consideration. In interbody anaesthesia techniques, in my opinion, the orientation of the needle with respect to the affected region can hydro dissect more in one direction (caudal) or the other (cranial). I would be very curious to know your experience in this regard and share this discussion. Somehow, I think it might be interesting for the paper as well.
Line 308-314: Do you think that, at least in part, it may also be useful for possible VBO, TECALBO or a simple hematoma of the auricle? and more broadly of the ear?
Conclusions
Well written
Author Response
Comment 1: Materials, methods and results are fairly well described. A few more efforts of completeness on the CT part (both for materials and methods as well as for results) and on the presentation of statistics would be appreciated (it does not compromise the quality of the paper).
Response 1: Thank you for your comments. We have reviewed the text together with our radiologist colleague and we don’t see how to further improve the explanation regarding the CT scan. If you have any specific suggestions, we would be happy to review it again. As for the statistics, we have modified the text, hoping it will now be easier to read.
Comment 2: The discussions are very interesting, comprehensive but not so long that the reader loses attention. Below you will find a word file with my brief remarks. They are more of my personal curiosities/thoughts to share with you and which I hope will give you new insights that you can also include in the discussions if you think it makes sense.
Response 2: Thank you very much for your conclusion. We will make an effort to address your personal curiosities regarding the study in our upcoming responses.
Comment 3: Line 25-27: I suggest dividing the period into two sentences. You speak about human medicine before and then veterinary medicine in the same sentence. In this way I think it is clearer.
Response 3: Thank you for your insightful suggestion to divide the sentence into two. We appreciate your attention to detail, which has helped us enhance the clarity of our manuscript.
Comment 4: Line 67-71: I suggest dividing the sentence into two period. Like this the sentence is long
Response 4: Thank you for your valuable feedback regarding the long sentence. Upon reviewing the article, we understood that the lengthy sentence referred to was from lines 69 to 72. We have divided it into two parts to simplify its reading and enhance clarity.
Comment 5: Line 143: “the platysma was removed”… I suggest to insert “platysma muscle…
Response 5: Thank you for your input. We have added the word 'muscle' to facilitate comprehension.
Comment 6: Line 149-150: Do you find certain anatomical differences between cadavers? For example, different breeds or dolichomorph vs mesomorph vs brachimorph dogs?
Response 6: In our opinion, we did not observe anatomical differences among the cadavers beyond the size variations associated with the animals. It is possible that, due to the majority being mixed-breed dogs, specific modifications were not evident. However, we cannot discount the possibility that certain breeds, such as brachycephalic dogs, may exhibit anatomical differences that could influence the approach.
Comment 7: Line 275-276: In addition to your remarks on dissection (which I believe to be correct) I have another observation. It would be interesting to know if the solution arrived at C6 what type of dog breed it was. And it would also be interesting to know if the orientation of the spinal needle. Actually this is my personal impression, a consideration. In interbody anaesthesia techniques, in my opinion, the orientation of the needle with respect to the affected region can hydro dissect more in one direction (caudal) or the other (cranial). I would be very curious to know your experience in this regard and share this discussion. Somehow, I think it might be interesting for the paper as well.
Response 7: Thank you very much for your reasoning regarding these topics. Concerning the animal in which the contrast reached C6, it was a mixed-breed dog of intermediate weight, so we cannot suggest that this could be attributed to any specific anatomical characteristic of the breed.
Regarding the orientation of the spinal needle, the author who performed the injections agrees with your opinion that it may affect the variation in hydrodissection, either more cranially or caudally. Therefore, the orientation used was craniocaudal. We did not perform injections with a caudocranial orientation to avoid introducing a potential bias in the results.
Comment 8: Line 308-314: Do you think that, at least in part, it may also be useful for possible VBO, TECALBO or a simple hematoma of the auricle? and more broadly of the ear?
Response 8: Thank you for your interest regarding this topic. Although we have not utilized the block in cases of VBO, we have successfully employed it in TECABO with excellent results. We only observed mild nociceptive stimulus, which responded well to analgesic rescue, at the time of the bulla osteotomy, resulting in an excellent immediate postoperative outcome.
Regarding aural hematomas, we have not evaluated this block; however, it can be assumed that it may not be effective since the skin of the inner surface of the auricle is innervated only to a small extent by the greater occipital nerve, with the remainder innervated by different nerves unrelated to the cervical branches.
Reviewer 4 Report
Comments and Suggestions for Authors
Dear authors,
Thank you so much for this well done, pleasant to read manuscript. I just have a few comments to add.
I am curious if based on the study design you were able to exclude the situation when bilateral local anaesthetic distribution might block both phrenic nerves after a single site inoculation. In other words, if I would like to do this block for a left side arytenoid lateralization, is the patient of risk to stop breathing?
A big limitation of your study is that unfortunately it was not blinded. However, you acknowledge this in the discussions section.
It might be useful to add the clinical reasons for this type of block in the Summary section.
This manuscript mentions nothing about the polymorphism of dog breeds which definitely will have an impact on your anatomical and US images. What were the breeds of your subjects? Please add this information in the MM section.
Line 61-62: different font size used
Line 132: please mention the details regarding the type of US gel used (name, company and country of production)
Line 133: please mention the details regarding the type of US probe ( company and country of production)
Line 135: If the probe was positioned under the transverse process, maybe you should say that the approach is not lateral but rather latero-ventral. Just a suggestion.
Line 159: I would rather change the word "located" with "placed"
Line 160: I would say "Based on phase I"
Line 167: use superscript for "-1". the same at line 362
Line 169: I would rather say "the same person" as the patient was not under anaesthesia.
Line 177: "two persons" sounds better from my point of view. The same for line 185.
Line 214: I would also suggest you to provide a transverse/axial drawn image on the angle at which the probe faces the sagittal plane. Or mention in the text that the probe was placed perpendicular to the table or skin, if this was the case.
Line 273: I would not express the results in percentages as the number of cases was small. You should better replace 100% to "all" or "6 out of 6"
Line 285-286: In dogs, you need to consider the breed too as there is great polymorphism between breeds.
Line 287-288: please do not repeat detailed data already presented in the results section.
Line 347-348: Does your study exclude the situation where one side inoculation with the technique described will not affect both phrenic nerves.
I am looking forward to see your reply.
Author Response
Comment 1: I am curious if based on the study design you were able to exclude the situation when bilateral local anaesthetic distribution might block both phrenic nerves after a single site inoculation. In other words, if I would like to do this block for a left side arytenoid lateralization, is the patient of risk to stop breathing?
Response 1: Thank you for your comment and the opportunity to clarify our findings. Based on the results of this study, we believe that phrenic nerve blockade would be unusual. In our clinical experience, we have applied this block without observing any alterations in diaphragmatic movement. Additionally, we have performed the block bilaterally on several occasions, confirming diaphragmatic movement using ultrasound. Furthermore, we have published a case of surgery for laryngeal hemiplegia in which a unilateral block was employed without compromising diaphragmatic function. These findings support that the phrenic nerve block, as we perform it, presents a low risk of impacting the diaphragm; therefore, respiratory arrest is not expected following this procedure. However, clinical studies will be necessary to further evaluate the safety and effectiveness of this block in dogs.
Comment 2: A big limitation of your study is that unfortunately it was not blinded. However, you acknowledge this in the discussions section.
Response 2: Thank you for your comment. We had previously acknowledged the limitation of not performing the procedure blindly in the text, as we considered it essential to provide a transparent account of our methodology.
Comment 3: It might be useful to add the clinical reasons for this type of block in the Summary section.
Response 3: Thank you very much for the suggestion. We have added the clinical indications to the Simple Summary to provide the reader with more general information.
Comment 4: This manuscript mentions nothing about the polymorphism of dog breeds which definitely will have an impact on your anatomical and US images. What were the breeds of your subjects? Please add this information in the MM section.
Response 4: Thank you for focusing on this topic. We have added information regarding the breeds used in our study, as it is an interesting aspect to consider. However, we did not observe any differences, as the cadavers used were mostly of mixed-breed origin.
Comment 5: Line 61-62: different font size used.
Response 5: Thank you for identifying this font size change; we have now corrected it.
Comment 6: Line 132: please mention the details regarding the type of US gel used (name, company and country of production).
Response 6: Thank you very much for your suggestion, we have added the information about the gel used.
Comment 7: Line 133: please mention the details regarding the type of US probe (company and country of production).
Response 7: Thank you for your suggestion, we have added the information about the ultrasound probe used.
Comment 8: Line 135: If the probe was positioned under the transverse process, maybe you should say that the approach is not lateral but rather latero-ventral. Just a suggestion.
Response 8: Thank you for the suggestion; we have changed the term 'lateral' to 'latero-ventral' to more accurately define the area where the scan was performed.
Comment 9: Line 159: I would rather change the word "located" with "placed"
Response 9: Thank you for your collaboration; 'placed' is the most suitable word for the sentence.
Comment 10: Line 160: I would say "Based on phase I".
Response 10: Thank you for your input; we have made the corresponding modification.
Comment 11: Line 167: use superscript for "-1". the same at line 362.
Response 11: Thank you for your input; we have made the corresponding modification.
Comment 12: Line 169: I would rather say "the same person" as the patient was not under anaesthesia.
Response 12: Thank you for your input; we have made the corresponding modification.
Comment 13: Line 177: "two persons" sounds better from my point of view. The same for line 185.
Response 13: Thank you for your input; we have made the corresponding modification.
Comment 14: Line 214: I would also suggest you to provide a transverse/axial drawn image on the angle at which the probe faces the sagittal plane. Or mention in the text that the probe was placed perpendicular to the table or skin, if this was the case.
Response 14: Thank you for your suggestion; we have added in the text that the probe is placed perpendicular to the skin. We prefer not to add a new drawing to avoid overcrowding the figure.
Comment 15: Line 273: I would not express the results in percentages as the number of cases was small. You should better replace 100% to "all" or "6 out of 6"
Response 15: Thank you for your input; we have made the corresponding modification.
Comment 16: Line 285-286: In dogs, you need to consider the breed too as there is great polymorphism between breeds.
Response 16: Thank you for your observation. In addition to specifying the breeds of the animals in the Materials and Methods section, we have incorporated this consideration into the Discussion.
Comment 17: Line 287-288: please do not repeat detailed data already presented in the results section.
Response 17: Thank you for your suggestion to avoid repeating results. We have revised the text accordingly to enhance clarity and conciseness.
Comment 18: Line 347-348: Does your study exclude the situation where one side inoculation with the technique described will not affect both phrenic nerves.
Response 18: Thank you for your suggestion regarding the phrenic nerve. We have added this topic to the discussion. Based on our results, bilateral phrenic nerve block would be unlikely, though future clinical studies will need to confirm the efficacy and safety of this technique.
Round 2
Reviewer 4 Report
Comments and Suggestions for Authors
Dear authors,
you did a great job with these improvements. Please double-check the font size throughout the text as in the Summary section you used two sizes.